# Cellular PSMB4 Protein Suppresses Influenza A Virus Replication through Targeting NS1 Protein

**DOI:** 10.3390/v14102277

**Published:** 2022-10-17

**Authors:** Chee-Hing Yang, Che-Fang Hsu, Xiang-Qing Lai, Yu-Ru Chan, Hui-Chun Li, Shih-Yen Lo

**Affiliations:** 1Department of Laboratory Medicine and Biotechnology, Tzu Chi University, No. 701, Section 3, Chung Yang Road, Hualien 97004, Taiwan; 2Center for Prevention and Therapy of Gynecological Cancers, Department of Medical Research, Hualien Tzu Chi Hospital, Buddhist Tzu Chi Medical Foundation, Hualien 97004, Taiwan; 3Department of Biochemistry, Tzu Chi University, Hualien 97004, Taiwan; 4Department of Laboratory Medicine, Buddhist Tzu Chi General Hospital, Hualien 97004, Taiwan

**Keywords:** influenza A virus, NS1 protein, PSMB4, protein degradation, MG132

## Abstract

The nonstructural protein 1 (NS1) of influenza A virus (IAV) possesses multiple functions, such as the inhibition of the host antiviral immune responses, to facilitate viral infection. To search for cellular proteins interacting with the IAV NS1 protein, the yeast two-hybrid system was adopted. Proteasome family member PSMB4 (proteasome subunit beta type 4) was found to interact with the NS1 protein in this screening experiment. The binding domains of these two proteins were also determined using this system. The physical interactions between the NS1 and cellular PSMB4 proteins were further confirmed by co-immunoprecipitation assay and confocal microscopy in mammalian cells. Neither transiently nor stably expressed NS1 protein affected the PSMB4 expression in cells. In contrast, PSMB4 reduced the NS1 protein expression level, especially in the presence of MG132. As expected, the functions of the NS1 protein, such as inhibition of interferon activity and enhancement of transient gene expression, were suppressed by PSMB4. PSMB4 knockdown enhances IAV replication, while its overexpression attenuates IAV replication. Thus, the results of this study suggest that the cellular PSMB4 protein interacts with and possibly facilitates the degradation of the NS1 protein, which in turn suppresses IAV replication.

## 1. Introduction

Influenza A virus (IAV) still poses a serious threat to human lives and global economies (https://www.who.int/en/news-room/fact-sheets/detail/influenza-(seasonal), accessed on 6 September 2022). IAV, a member of the family of Orthomyxoviridae, contains eight segmented genomic RNAs of negative polarity. At least 15 different viral proteins are encoded in the IAV genome: hemagglutinin (HA), neuraminidase (NA), matrix protein M1, M2, PA, PB1, PB2, nucleoprotein (NP), non-structural protein 1 (NS1), PB1-F2, PA-X, PA-N155, PA-N182, N40 and NEP. The subtyping of IAV is based on the sequences of the two major surface glycoproteins (i.e., HA and NA). At present, 18 subtypes of HA (H1 to H18) and 11 subtypes of NA (N1 to N11) have been identified from different animal hosts [1].

Each IAV protein plays specific roles in the virus lifecycle. Among these viral proteins, NS1 is a multifunctional protein that plays a key role in IAV replication, virulence and inhibition of the cellular antiviral immune response [2]. The length of the NS1 protein varies, typically 202–237 amino acids (a.a.) depending on the viral strain. NS1 is most often a 230 a.a. protein, e.g., WSN33 (H1N1). The NS1 protein can be divided into two functional domains: an N-terminal RNA-binding domain (RBD, a.a. 1–73 in WSN33) and a C-terminal effector domain (ED, a.a. 74–230 in WSN33) [3,4,5]. The RBD could bind to both single-stranded and double-stranded RNAs, thereby blocking their recognition by the retinoic acid-induced gene and thus inhibiting interferon (IFN) expression [6]. The ED could interact with various host cell proteins, e.g., by binding to CPSF30 to block the processing of cellular mRNAs [7]. The major role of the NS1 protein is the inhibition of both IFN and IFN-stimulated proteins [8]. Indeed, NS1 could antagonize the IFN signaling response by inhibiting IRF3 activation [9] or by regulating other cellular factors, such as phosphoinositide 3 kinase (PI3K) activity [10,11].

Many cellular proteins interacting with the NS1 protein were identified by various approaches, e.g., immunoprecipitation followed by SDS-PAGE coupled with liquid chromatography–tandem mass spectrometry (GeLC–MS/MS) [12], the yeast two-hybrid system [13] or bioinformatics [14]. To explore the functions of the IAV NS1 protein, a yeast two-hybrid screening experiment was conducted in this study to identify cellular proteins that may interact with the NS1 protein. Cellular proteasome subunit beta type 4 (PSMB4), a member of the proteasome family, was found to interact with the NS1 protein in this experiment. Expression of the PSMB4 gene was detected in the endometrium epithelium, the epithelium of the nasopharynx, the right adrenal gland and many other tissues (https://bgee.org/gene/ENSG00000159377, accessed on 6 October 2022). Our studies further showed that PSMB4 knockdown enhances IAV replication, while its overexpression attenuates IAV replication. Thus, PSMB4 serves as a restriction factor for IAV infection.

## 2. Materials and Methods

### 2.1. Yeast Two-Hybrid System

The yeast two-hybrid screening system was purchased from Clontech Laboratories (Mountain View, CA, USA). The screening procedures were performed following the manufacturer’s instructions and our previous protocols [15]. The IAV NS1 gene was cloned into the pBD-GAL4 Cam phagemid vector (Agilent Technology, Palo Alto, CA, USA). A human fetal liver cDNA library (HL4029AH, cloned into prey vector pACT2, Clontech, Mountain View, CA, USA) was used for this screening. *Saccharomyces cerevisiae* YRG-2 strain transformed with plasmids was selected in -W-L-H + G plates (medium containing glucose but not Trp, Leu and His). Positive colonies growing on -W-H-L + G plates were streaked on -W-L-H + G + 3AT (100 mM) for further selection. Plasmid DNA extracted from the individual yeast clones grown in -W-L-H + G + 3AT selection plates was transformed into *E. coli* (DH5α) competent cells separately. Then, cDNA inserts in pACT2 were extracted from these bacteria grown in ampicillin selection plates and identified by sequencing [15].

### 2.2. Plasmid Construction and DNA Transfection

The expression plasmids used in this study were constructed using standard protocols through the amplification of gene fragments by polymerase chain reaction (PCR), as described previously [16]. The primer sequences used for PCR are listed in Table 1. All expression plasmids were verified by DNA sequencing. The commercially available polyethylenimine (PEI) (linear, MW 25,000) was used to transfect DNA into cells (Polysciences Inc., Warrington, PA, USA).

### 2.3. Cell Culture

A549 cells (http://rnai.genmed.sinica.edu.tw, accessed on 6 September 2022) were cultured in Ham’s F12 Nutrient Mixture with Kaighn’s Modification (F12K, Sigma, St. Louis, MO, USA), containing 2.5 g/L NaHCO_3_, 10% fetal bovine serum (FBS), 100 U/mL penicillin and 100 μg/mL streptomycin (P + S) (Gibco, Waltham, MA, USA). Madin–Darby canine kidney (MDCK) [16] and 293T cells (http://rnai.genmed.sinica.edu.tw, accessed on 6 September 2022) were cultured in Dulbecco’s modified Eagle’s medium (DMEM) containing 10% FBS, 1% glutamine, 1 mM sodium pyruvate, P + S. HeLa cells were cultured in RPMI medium 1640 (Gibco) containing 2 g/L NaHCO_3_, 10% FBS, P + S. All cultured cells were maintained at 37 °C with 5% CO_2_.

### 2.4. Co-Immunoprecipitation Assay

A549 cells (2 × 10^6^) were harvested 48 h after transfection with the expression plasmids pcDNA3-6HA-PSMB4 and/or pcDNA3.1-NS1, washed three times in phosphate-buffered saline (PBS) and lysed in RIPA buffer (150 mM NaCl, 1% NP40, 0.5% deoxycholic acid, 0.1% SDS and 50 mM Tris, pH 7.5). After centrifugation for 5 min at full speed, the supernatant was used for further analysis. In each experiment, 10% of the supernatant was used for expression analysis (input) by Western blotting directly, while 90% of the supernatant was used for the co-immunoprecipitation assay. The supernatant was incubated with anti-V5 at 4 °C with shaking overnight. The protein A Mag Sepharose bead (GE Healthcare, Chicago, IL, USA) was added to pull down the antigen–antibody complexes. The immunoprecipitated pellets were then treated at 100 °C in the sample buffer (67.5 mM Tris–HCl (pH 6.8), 5% 2-mercaptoethanol, 3% SDS, 0.1% bromophenol blue and 10% glycerol) for 10 min, followed by the assays of SDS-PAGE and Western blotting.

### 2.5. Western Blotting Analysis

In general, cell lysates for SDS-PAGE were collected 48 h after expressing plasmid transfection. If necessary, cells were treated with proteasome inhibitor MG132 at 10 uM for 12 h. After electrophoresis, proteins from the SDS-PAGE gel were transferred to PVDF paper (Pall Corporation, New York, NY, USA). All procedures were then carried out at room temperature [17]. The primary and secondary antibodies used in this study are listed in Table 2. After the assay, the relative amounts of different proteins were quantified using the software “Quantity One” (Bio-Rad, Hercules, CA, USA).

### 2.6. Confocal Analysis

Approximately 2.5 × 10^5^ A549 cells were seeded in 35 mm culture dishes. After overnight incubation, the cells were transfected with plasmids (pcDNA3-myc-PSMB4 and/or pcDNA3.1-NS1). The exogenously expressed proteins in the cells were analyzed 48 h after transfection, following our previous procedures [18]. Anti-myc mouse monoclonal antibody was used, followed by RITC-conjugated anti-mouse IgG antibody to detect PSMB4 protein and FITC-conjugated anti-V5 monoclonal antibody to detect NS1 protein. DNA was stained by DAPI (Merck, Darmstadt, Germany) for nucleus localization. Confocal analysis was done with Leica TCS SP2 (Leica, Wetzlar, Germany).

### 2.7. Virus Infection

A549 cells were used for infection with IAV WSN33 (H1N1) in this study. MDCK cells were used for IAV amplification and plaque assay, following previously published procedures [16].

### 2.8. Knockdown Experiments and Stable Cell Establishment

The shRNA knockdown experiments and establishment of stable cells were conducted using the lentiviral expression system (http://rnai.genmed.sinica.edu.tw, accessed on 6 September 2022), following the manufacturer’s instructions. Reagents (shRNA clones for gene knockdown and plasmids for gene overexpression) were obtained from the National RNAi Core Facility located at the Institute of Molecular Biology/Genomic Research Center, Academia Sinica, Taiwan.

### 2.9. Luciferase Assay

The Dual-Glo Luciferase Assay System (Promega, Madison, WI, USA) was used for the luciferase assays, following the manufacturer’s instructions. In each experiment, triplicate samples were analyzed.

### 2.10. Assay for Interferon Activity

pISRE-Luc, kindly provided by Prof. R.L. Kuo (Research Center for Emerging Viral Infections, College of Medicine, Chang Gung University, Taoyuan, Taiwan), is a luciferase reporter plasmid under the control of the interferon-stimulated response element (ISRE) promoter. The reporter plasmid pISRE-Luc was co-transfected with NS1-V5 and/or 6xHA-PSMB4 expression plasmids into HEK293T cells. pCDNA4-RLuc was used for transfection efficiency control. Two hours after transfection, 20 ug/mL polyI:C was added to stimulate interferon expression. After further incubation for 24 h, the cells were washed with PBS. Then, 1× PassiveLysis Buffer (Promega, Madison, WI, USA) was used to lyse the cells for the dual luciferase assay.

### 2.11. Statistical Analysis

Experiments involving plaque and luciferase assays were performed at least twice in triplicate. Data were analyzed using the Student *t* test, and *p* < 0.05 was considered statistically significant (* *p* < 0.05, ** *p* < 0.01, *** *p* < 0.001).

## 3. Results

### 3.1. Cellular PSMB4 Was Found to Interact with IAV NS1 Protein Using Yeast Two-Hybrid System

To elucidate the multiple functions of the IAV NS1 protein and search for cellular factors interacting with the NS1 protein, a yeast two-hybrid screening assay was conducted. The IAV WSN33 (H1N1) NS1 full-length protein was used as bait for screening possible interacting cellular proteins. IAV will not infect hepatocytes. To determine the specificity of this screening system, a liver cDNA library was used for this screening. After screening, of the 56 yeast clones harboring cDNAs encoding potential NS1 binding proteins, 25 of them contained the coding region of proteasome subunit beta type 4 (PSMB4) with different lengths, with 21 clones encoding ribosomal protein L41 (RPL41), and others with inserts of un-identified genes or no insert. Thus, this screening is rather specific, because no liver-specific genes (e.g., albumin) were identified in this screening system. We could not detect RPL41 protein in Western blotting analysis, possibly because it is a basic (positively charged) peptide consisting of only 25 a.a. [19]. The interaction of IAV NS1 and cellular PSMB4 proteins was further characterized in this study.

To further identify the interactive regions of PSMB4 and NS1 proteins, deletion-mapping experiments were performed. The PSMB4 protein contains a propeptide (1–45), mature protein (46–250) and C-terminal region (251–264) [20]. None of the 25 clones with PSMB4 coding sequences contained the first 72 residues of this protein. Thus, the full-length, N-terminal region (1–72) and C-terminal region (73–264) of PSMB4 were individually cloned and analyzed in this study. As shown in Figure 1, the C-terminal ED of the NS1 protein but not the N-terminal RBD could interact with PSMB4. Similarly, the C-terminal region of the PSMB4 protein but not the N-terminal region could bind to the ED of NS1 (Figure 1). Thus, the C-terminal region of PSMB4 (73–264) interacts with the C-terminal ED of the NS1 protein.

### 3.2. Interactions between PSMB4 and NS1 in Mammalian Cells

We also performed a co-immunoprecipitation experiment to test whether NS1 and PSMB4 could bind to each other in cells. The V5-tagged full-length NS1 protein and the HA-tagged PSMB4 protein were co-expressed in A549 cells by transient transfection. After transfection, cell lysates were immunoprecipitated with the anti-V5 antibody, followed by Western blotting using the anti-HA or anti-V5 antibodies. As shown in Figure 2, the HA-tagged PSMB4 protein could be immunoprecipitated by the anti-V5 antibody in the presence but not in the absence of the NS1 protein (bottom panel). This result further confirmed that PSMB4 and NS1 could bind to each other.

If the NS1 and PSMB4 proteins bind to each other in cells, they should be at least partially co-localized in cells. Confocal microscopy was used to examine the subcellular localization of these two proteins. Plasmids expressing the full-length NS1 protein with the V5 tag and the PSMB4 protein with the myc tag were transfected individually or together into A549 cells. When these two proteins were expressed individually, similar to previous reports [5,21,22], the NS1 protein was located predominately in the nucleus and the PSMB4 protein in the cytoplasm (Figure 3A). However, a significant portion of the NS1 protein was co-localized with PSMB4 in the cytoplasm when these two proteins were co-expressed in cells (Figure 3B). These results indicated that NS1 and PSMB4 indeed physically interact with each other in cells.

### 3.3. Neither IAV Infection nor NS1 Protein Modulates PSMB4 Expression Significantly

To determine whether IAV modulates the PSMB4 expression to benefit its replication, the endogenous PSMB4 protein level was detected after IAV infection (Figure 4A). The PSMB4 protein expression fluctuated after IAV infection. However, IAV did not affect the PSMB4 expression significantly. To determine whether the NS1 protein affects the PSMB4 protein expression level due to their physical interactions in cells, NS1 was exogenously expressed in cells using transient or stable expression. However, the NS1 protein did not affect the PSMB4 amount in either transient (Figure 4B) or stable expression (Figure 4C).

### 3.4. PSMB4 Reduces NS1 Protein Expression

To determine whether PSMB4 protein affects the NS1 protein expression due to their physical interactions in cells, expression plasmids encoding PSMB4 with HA tag and NS1 with V5 tag (or control protein eGFP) were transiently co-transfected in the cells. Under this condition, PSMB4 reduced the NS1 protein amount but not that of the control eGFP protein dose-dependently (Figure 5A). PSMB4 is a β subunit of the 20S core proteasome, mediating the degradation of damaged or unneeded cellular proteins [23]. To determine whether PSMB4 reduces the NS1 protein amount through the proteasome pathway, proteasome inhibitor MG132 was added. MG132 did prevent the degradation of eGFP, IAV M1 protein (also served as a control) and PSMB4 proteins (Figure 5B). Surprisingly, the reduction in the NS1 protein level by PSMB4 was exaggerated in the presence of MG132 (Figure 5B).

### 3.5. PSMB4 Degraded NS1 in MG132-Independent Pathway

It is possible that MG132 prevents the degradation of PSMB4, which in turn facilitates NS1’s degradation through an MG132-independent pathway. To test this hypothesis, expression plasmids encoding NS1 with the V5 tag or eGFP were transfected separately in cells with or without the treatment of MG132 (Figure 5C). The results showed that MG132 did prevent the degradation of eGFP but facilitated the degradation of the NS1 protein. To determine the role of PSMB4 in the NS1 degradation pathway, cells with shRNAs targeting PSMB4 were established in A549 cells. The knockdown efficiency of various shRNAs against PSMB4 was analyzed by Western blotting (Figure 5D). Cells with sh-PSMB4 (4) were used for the following experiments due to the better knockdown efficiency. Expression plasmids encoding NS1 with the V5 tag or eGFP were transfected separately in the cells with sh-PSMB4 or with control sh-scramble in the absence or presence of MG132 (Figure 5E). MG132 inhibited the degradation of eGFP in both cells with sh-scramble and with sh-PSMB4. In contrast, NS1 degradation was facilitated by MG132 in cells with sh-scramble but not with sh-PSMB4. Thus, NS1 degradation by PSMB4 occurs through an MG132-independent pathway.

### 3.6. PSMB4 Suppressed NS1 Functions

It is reasonable to assume that PSMB4 regulates the functions of NS1 through facilitating its degradation. The most well-known function of the NS1 protein is to inhibit IFN production and activity [4]. To determine whether PSMB4 affects the inhibition of IFN activity by NS1, the reporter under the control of the interferon-stimulated response element (ISRE) promoter was used to detect the IFN production induced by the treatment of poly I/C (Figure 6A). As expected, NS1 inhibited the IFN activity induced by poly I/C treatment. Moreover, PSMB4 overcame the NS1-induced inhibition of IFN activity.

Translational enhancement of transient gene expression is another function of the NS1 protein [24,25]. To determine whether PSMB4 regulates the translational enhancement of transient expression by NS1, luciferase was used as a reporter (Figure 6B). In the absence of NS1, the luciferase expression was not affected by PSMB4 (Figure 6B). However, in the presence of NS1, the luciferase expression was enhanced, which could be suppressed by PSMB4 (Figure 6C). These experiments were conducted in HeLa cells, and similar results were obtained in A549 cells. Experiments conducted with eGFP as a reporter obtained the same results: NS1 enhanced the eGFP expression and PSMB4 suppressed this enhancement (Figure 6D). To further verify these results, HeLa cells stably expressing NS1 or a control protein (luciferase) were established (Figure 6E). Expression of eGFP was elevated in HeLa cells stably expressing NS1, compared to that in cells stably expressing the control luciferase protein (1.87:1). PSMB4 suppressed the eGFP expression in cells stably expressing NS1 dose-dependently but not in cells stably expressing the control luciferase protein (Figure 6E). Thus, these results indicate that PSMB4 could suppress the enhancement in transient gene expression caused by the NS1 protein.

### 3.7. PSMB4 Could Inhibit IAV Replication

NS1 is critical for IAV replication, and IAV without NS1 could only replicate in IFN-deficient cells [26,27]. Thus, it is possible that PSMB4 inhibits IAV replication via its interaction with and suppression of the NS1 protein. To address this issue, A549 cells with PSMB4 knockdown or overexpression were established. IAV replication in the PSMB4 knockdown cells increased significantly, as shown by the increased expression of the M1 protein at either a low multiplicity of infection (MOI) of 0.01 or a high MOI of 3 (Figure 7A). Moreover, the extracellular IAV production was significantly elevated in cells with sh-PSMB4 at an MOI of either 0.01 or 3 (Figure 7B). In contrast, IAV replication (M1 protein level in Figure 7C) and extracellular IAV production (Figure 7D) in PSMB4-overexpressed cells were significantly reduced at either a low or high MOI. Therefore, PSMB4 knockdown enhanced IAV replication, while its overexpression attenuated IAV replication.

MG132 facilitated NS1 degradation (Figure 5C), possibly through preventing PSMB4 degradation (Figure 5B). Thus, MG132 should suppress IAV replication. Indeed, IAV production is reduced in the presence of MG132 (Figure 7E).

## 4. Discussion

The proteasome is responsible for the degradation of damaged or unneeded proteins in eukaryotic cells and thus controls many cellular processes, including protein homeostasis, signaling transduction, the cell cycle and the stress response [23]. Dysfunction of protein degradation by the proteasome is associated with diverse human diseases, such as cancer development and neurodegeneration [28]. PSMB4 is a β subunit of the 20S core proteasome (https://www.genecards.org/cgi-bin/carddisp.pl?gene=PSMB4&keywords=PSMB4, accessed on 6 September 2022). Though the biochemical functions of PSMB4 remain obscure, its involvement in several tumors and neuronal dysfunction has been documented [20,29,30,31,32,33]. Hepatitis B virus (HBV) X protein (HBx) was involved in the development of HBV-related hepatocellular carcinoma (HCC). Upregulation of PSMB4 was also detected in HBx-overexpressed cells [34] and transgenic mice with HCC [35]. Thus, PSMB4 should be important for the development of HBx-related HCC.

In this study, PSMB4 was found to interact with the IAV NS1 protein (Figure 1, Figure 2 and Figure 3). A previous report has also demonstrated that PSMB4 (HsN3) could bind to human immunodeficiency virus type 1 (HIV-1) Nef protein, which is important for viral pathogenicity, using a yeast two-hybrid system [36]. Moreover, downregulation of PSMB4 was detected after Norovirus infection [37]. Taken together, PSMB4 may be involved in the cellular defense mechanism against viral infections.

Though NS1 did not affect the cellular PSMB4 expression (Figure 4), it is still possible that NS1 would modulate proteasome activity through physical interaction with PSMB4. Further studies are required to prove this possibility. In this study, PSMB4 protein was shown to reduce the NS1 protein amount, especially in the presence of MG132 (Figure 5), suggesting that PSMB4 facilitates NS1 degradation in an MG132-independent pathway. To address this issue, cells with knockout or knockdown of PSMB4 are needed. We were unable to knock out a gene completely in HeLa or A549 cells using CRISPR previously [38]. Therefore, knockdown of PSMB4 in the cells was conducted. The sequence of the PSMB4 homolog in canines was not available online. Therefore, in order to knock down PSMB4, A549 and HeLa cells from humans, but not MDCK cells, were used in this study. Indeed, the functions of the NS1 protein, such as the inhibition of IFN production/activity and enhancement of gene transient expression, were suppressed by PSMB4 (Figure 6). Furthermore, PSMB4 knockdown enhances IAV replication, while its overexpression attenuates IAV replication (Figure 7). PSMB4 affects IAV replication more significantly with MOI 0.01 than with MOI 3 in either knockdown or overexpression conditions (Figure 7B,D), suggesting that more NS1 proteins from infections with a higher MOI counteract the PSMB4 protein.

The detailed mechanisms regarding the facilitation of NS1 reduction by PSMB4 in the presence of MG132 require further investigations. One possibility is that the degradation of PSMB4 but not that of the NS1 protein was prevented by MG132, although both proteins were degraded through the proteasome pathway. Another possibility is that the degradation of PSMB4 occurs through the proteasome while the degradation of NS1 does not occur through the proteasome, e.g., autophagy lysosomal pathway [39], though these two proteins did bind each other.

Proteasome inhibitor MG132 reduced NS1 expression (Figure 5C), possibly through its stabilization of the PSMB4 protein (Figure 5B). Thus, MG132 should inhibit IAV replication through the reduction of the NS1 protein. In agreement with a previous report conducted in MDCK cells [40], treatment of MG132 did suppress the IAV replication in A549 cells (Figure 7E). The effects of MG132 on IAV replication in this study and the previous one may not be exactly the same, because it is known that the IAV replication efficiency is different in these two types of cultured cells [41].

## 5. Conclusions

In summary, the results of our study suggest that cellular PSMB4 protein interacts with and possibly facilitates the degradation of the NS1 protein, which in turn suppresses IAV replication.

## Figures and Tables

**Figure 1 viruses-14-02277-f001:**
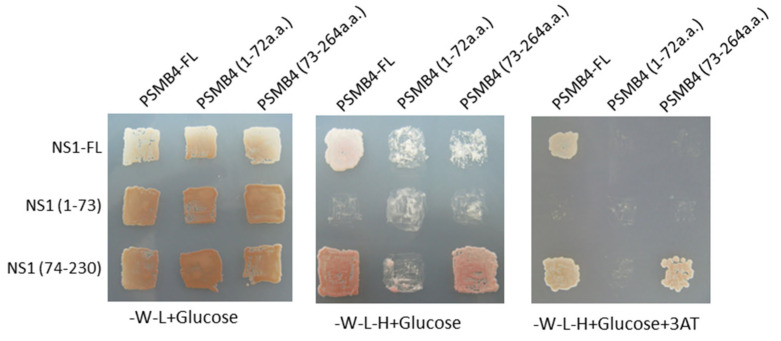
Cellular PSMB4 was found to interact with IAV NS1 protein. Growth of yeast transfected with plasmids as indicated on YNP without tryptophan and leucine (-W-L + Glucose, left panel), or on YNP without tryptophan, leucine and histidine in the absence of (-W-L-H + Glucose, middle panel) or presence of 100 mM 3AT (-W-L-H + Glucose + 3AT, right panel). 3AT, 3-amino-1,2,4-triazole, used as a competitive inhibitor of the His3 gene product.

**Figure 2 viruses-14-02277-f002:**
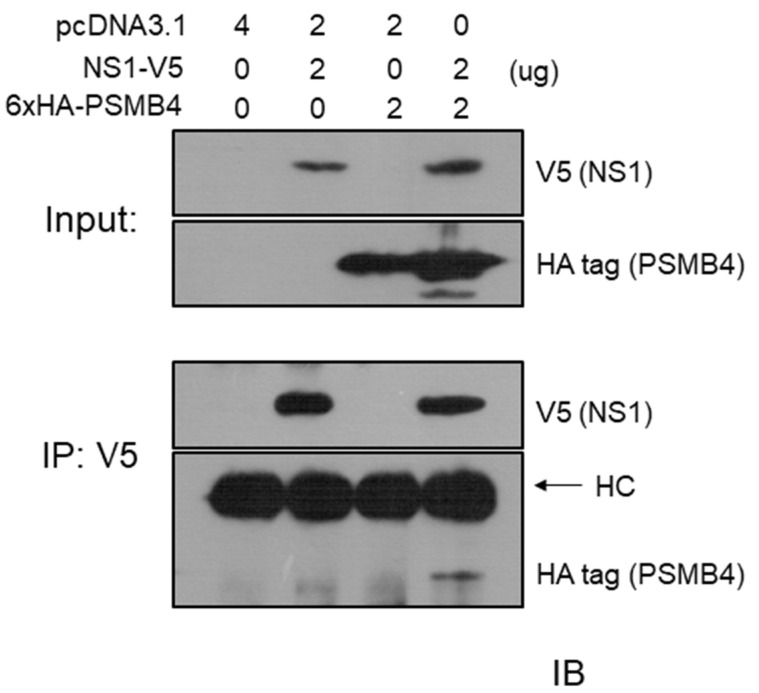
Co-immunoprecipitation experiments on IAV NS1 and PSMB4 proteins in A549 cells. A549 cells were transfected with a plasmid allowing expression of the PSMB4 protein with the HA tag, the NS1 protein with the V5 tag or co-transfected with these two plasmids. Cell lysates were analyzed directly (input) by Western blot (IB; immunoblot) or were immunoprecipitated (IP) with the anti-V5 antibody prior to Western blotting against V5 or HA tag.

**Figure 3 viruses-14-02277-f003:**
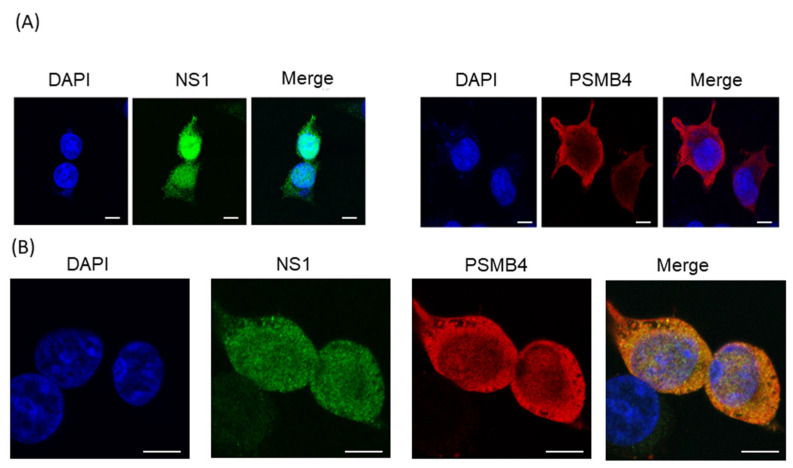
Confocal microscopy analysis of exogenous PSMB4 and IAV NS1 proteins in cultured cells. A549 cells were (**A**) transfected individually with the plasmid that expressed the NS1 protein with a V5 tag (left panel), the plasmid that expressed the PSMB4 protein with the myc tag (right panel) or (**B**) co-transfected with these two plasmids. Forty-eight hours after transfection, the cells were fixed and stained with mouse anti-myc, followed by RITC-conjugated anti-mouse antibody, and finally with FITC-conjugated anti-V5 antibodies. Blue, nucleus stained with DAPI; red, PSMB4 protein; green, NS1 protein; yellow, co-localization of NS1 and PSMB4 proteins. The scale bar represents 10 mm.

**Figure 4 viruses-14-02277-f004:**
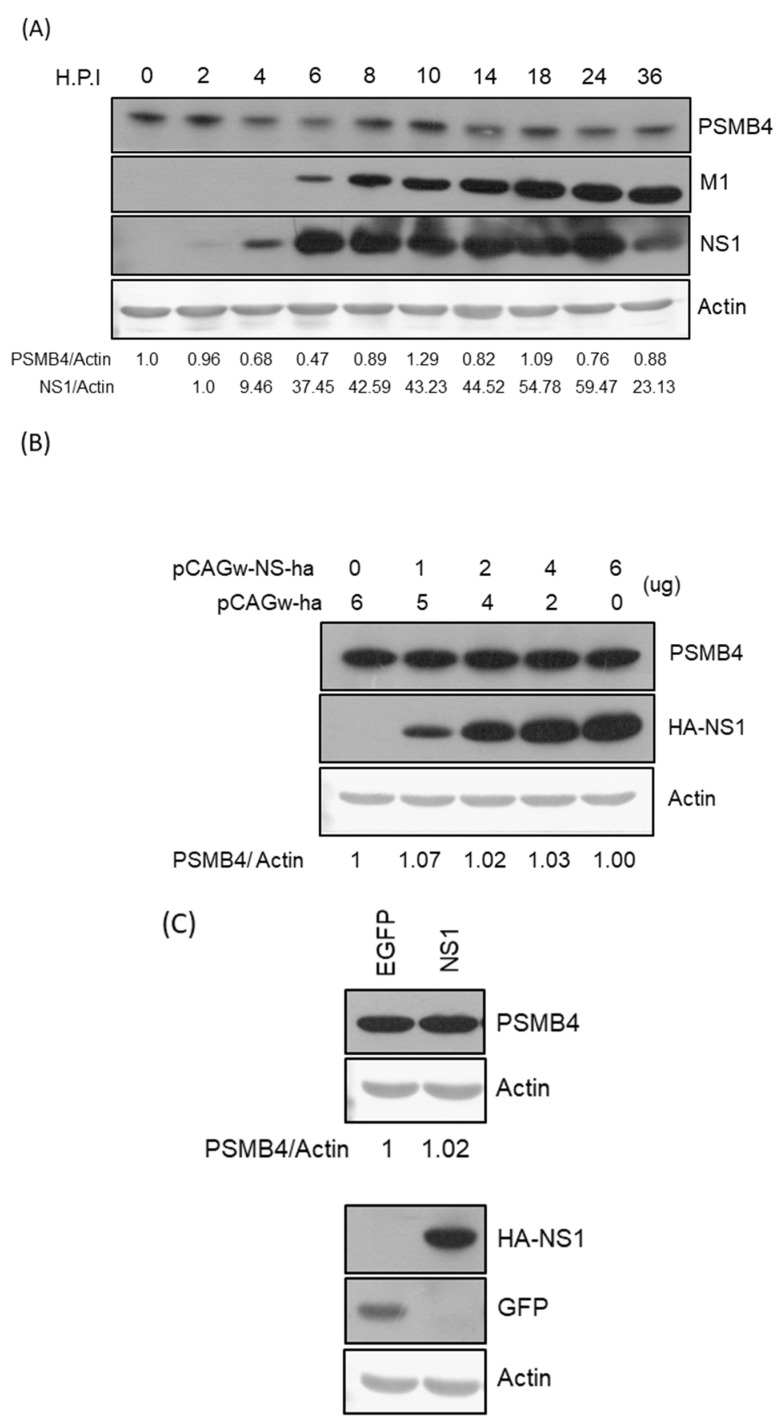
(**A**) Modulation of PSMB4 protein level by IAV infection. Cell lysates from A549 cells after infection with WSN 33 (H1N1) (MOI = 3) at different time points (hours post-infection, h.p.i.) as indicated were analyzed by SDS-PAGE and Western blotting against PSMB4 (upper panel), IAV M1 or NS1 protein (middle panels). Actin was used as the loading control (bottom panel). (**B**) Transiently expressed NS1 protein did not affect the endogenous PSMB4 protein level. Cell lysates from A549 cells transfected with plasmids encoding HA-tagged NS1 or empty vectors were analyzed by SDS-PAGE and Western blotting against PSMB4 (upper panel), HA tag (for exogenous NS1). Actin was used as the loading control (bottom panel). (**C**) Stably expressed NS1 protein did not affect the PSMB4 protein level. Cell lysates from A549 cells transduced with lentiviral vectors expressing GFP (as a control) or HA-tagged NS1 were analyzed by SDS-PAGE and Western blotting against different proteins as indicated. Only one representative result is shown in this figure. However, experiments were repeated at least two times and similar results were obtained.

**Figure 5 viruses-14-02277-f005:**
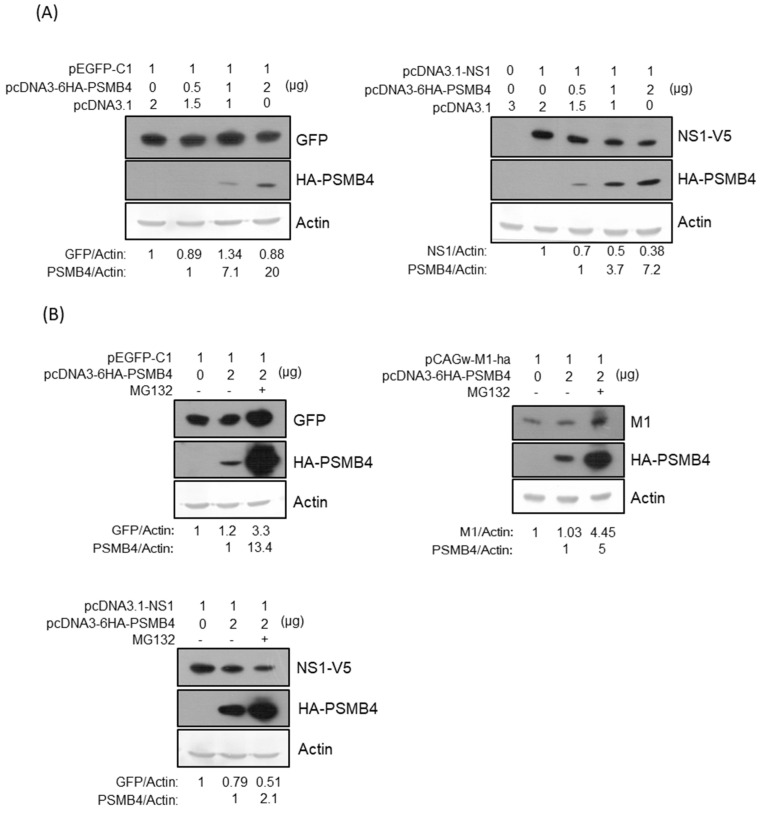
(**A**) Transiently co-transfected PSMB4 inhibited NS1 protein expression dose-dependently (right panel) but not GFP expression (left panel). Cell lysates from A549 cells transfected with different combinations of plasmids as indicated were analyzed by SDS-PAGE and Western blotting against GFP, against V5 (for exogenous NS1), HA tag (for exogenous PSMB4). Actin was used as the loading control (bottom panel). (**B**) Transiently co-transfected PSMB4 reduced NS1 protein expression (lower panel) but not that of GFP or IAV M1 protein (upper panel) in the presence of MG132. Cell lysates from A549 cells transfected with different combinations of plasmids as indicated and treated without or with 10 uM MG132 were analyzed by SDS-PAGE and Western blotting as indicated. (**C**) MG132 prevented GFP but facilitated NS1 protein degradation. Cell lysates from A549 cells transfected with plasmids encoding NS1 or GFP as indicated and treated with 10 uM MG132 or not were analyzed by SDS-PAGE and Western blotting as indicated. (**D**) Knockdown efficiency of various shRNAs targeting PSMB4. Cell lysates from A549 cells transduced with the lentiviral vectors expressing shRNAs targeting Luc (as the control) or PSMB4 (different clones) and selected with puromycin were analyzed by SDS-PAGE and Western blotting against PSMB4 or ERK2. (**E**) MG132 did not facilitate NS1 protein degradation in the cells with shRNA targeting PSMB4. Cell lysates from A549 cells with shRNAs targeting Luc or PSMB4 transfected with plasmids encoding NS1 or GFP in the presence of 10 uM MG132 or not were analyzed by SDS-PAGE and Western blotting.

**Figure 6 viruses-14-02277-f006:**
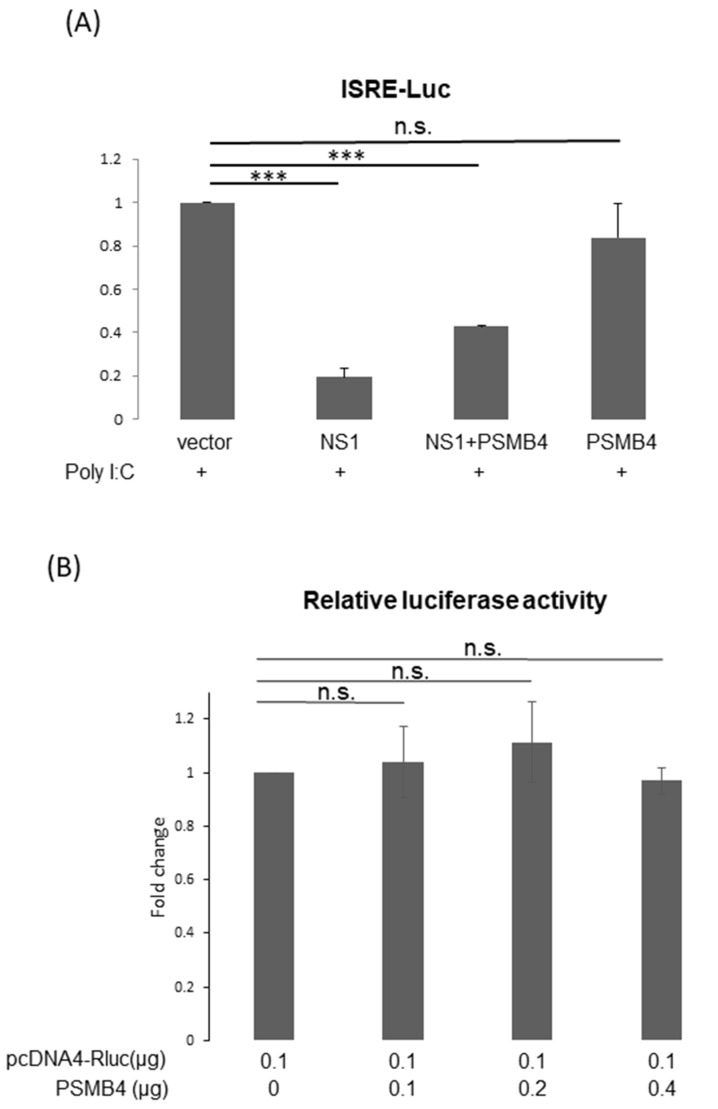
(**A**) PSMB4 overcame the inhibition of IFN activity by NS1. Cells were transfected with empty vector, expression plasmids for NS1, PSMB4 or both, as indicated. Poly I/C was added 2 h after transfection. Luciferase activities were analyzed 24 h after transfection. (**B**,**C**) PSMB4 did not modulate luciferase expression (**B**), while it suppressed the enhancement in luciferase expression caused by transiently expressed NS1. Luciferase activities were analyzed 48 h after cells were transfected with different combinations of plasmids as indicated. (**D**) PSMB4 suppressed the enhancement in GFP expression caused by transiently expressed NS1. Cell lysates from A549 cells transfected with different combinations of plasmids as indicated were analyzed by SDS-PAGE and Western blotting against GFP, V5 (for exogenous NS1), HA tag (for exogenous PSMB4), PSMB4 or actin. (**E**) PSMB4 suppressed the enhancement in GFP expression caused by stably expressed NS1. Cell lysates from HeLa cells stably expressing luciferase or NS1 protein transfected with different combinations of plasmids as indicated were analyzed by SDS-PAGE and Western blotting against GFP, HA tag (for exogenous PSMB4) or actin. *** *p* < 0.001.

**Figure 7 viruses-14-02277-f007:**
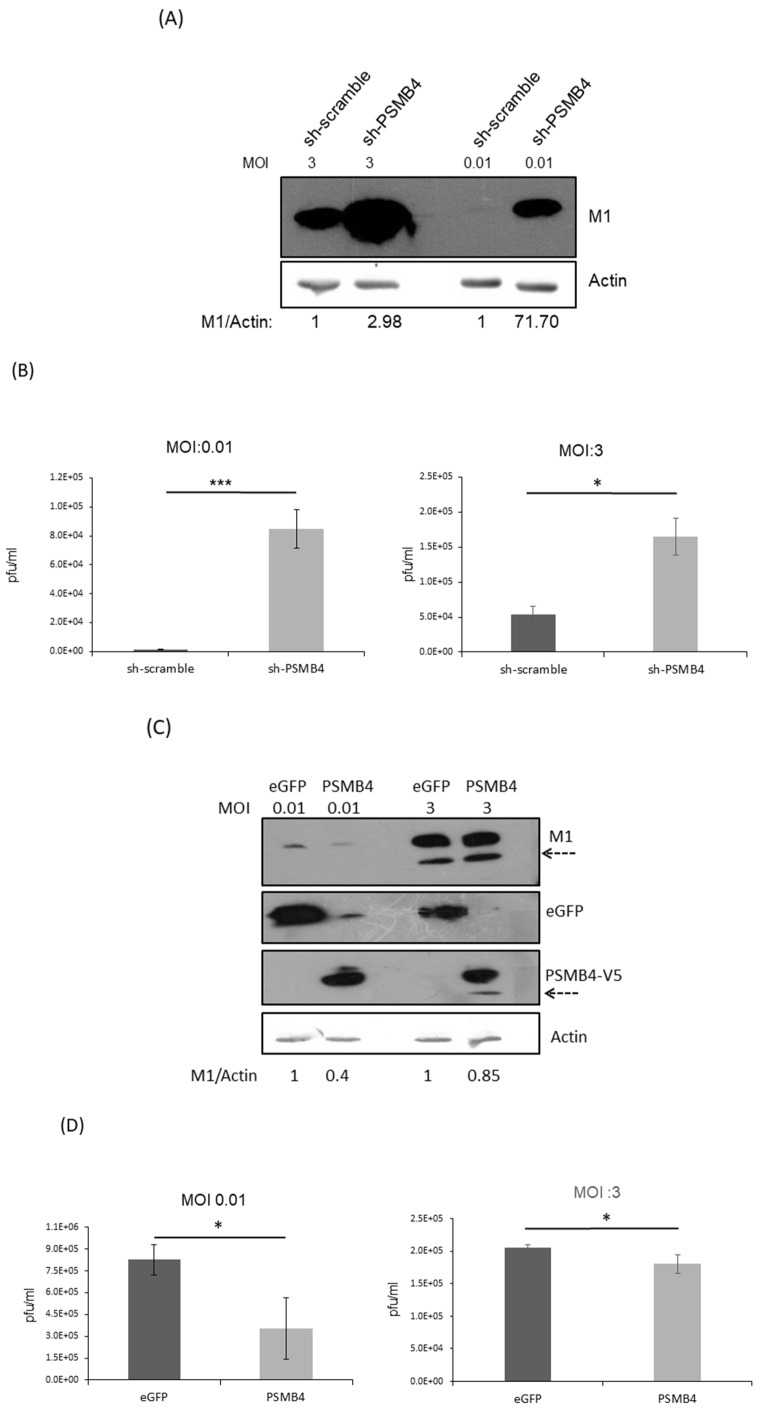
(**A**,**B**) Knockdown of PSMB4 enhanced IAV replication significantly. A549 cells transduced with sh-scramble (as the control) or sh-PSMB4 were infected with IAV (MOI = 3 or 0.01). Cell lysates were analyzed by SDS-PAGE and Western blotting against IAV M1 protein or actin (**A**) after 48 h of MOI 0.01 or 20 h of MOI = 3 infection, while secreted virions from the supernatant were quantitated by plaque assay (**B**) after 48 h of MOI 0.01 or 20 h of MOI = 3 infection. (**C**,**D**) Overexpression of PSMB4 suppressed IAV replication. A549 cells stably expressing GFP (as the control) or PSMB4 were infected with IAV (MOI = 0.01 or 3). Cell lysates were analyzed by SDS-PAGE and Western blotting against M1, GFP or V5 tag (for exogenous PSMB4) after 48 h of MOI 0.01 or 20 h of MOI = 3 infection (**C**), while secreted virions from the supernatant were quantitated by plaque assay after 48 h of MOI 0.01 or 20 h of MOI = 3 infection (**D**). The protein bands marked by the dotted arrow occasionally detected were probably the degraded M1 proteins. (**E**) Suppression of IAV replication by MG132. A549 cells were infected with IAV (MOI = 1) for 12 h. Then, cells were treated with 10 uM MG132 or not. Secreted virions from the supernatant were quantitated by plaque assay after another 12 h. * *p* < 0.05, *** *p* < 0.001.

**Table 1 viruses-14-02277-t001:** PCR primers used for plasmid construction.

Primer	Sequence
WSN33NS-S	5′-CG***GAATTC***ATGGATCCAAACACTGT-3′
WSN33NS-AS3	5′-TGC***TCTAGA***AACTTCTGACCTAATTGT-3′
WSN33NS-74S	5′-CG***GAATTC***ATGGATGAGGCACTCAAAAT-3′
WSN33NS-73AS	5′-TGC***TCTAGA***AGATTCTTCCTTCAGAAT-3′
BDNS1-AS	5′-CTATTAAACTTCTGACCTAATTGT-3′
BDNS1-73AS	5′-CTATTAAGATTCTTCCTTCAGAAT-3′
PSMB4-ADS	5′-CC***GAATTC***GCATGGAAGCGTTTTTGGG-3′
PSMB4-ADAS	5′-CCG***CTCGAG***TCATTCAAAGCCACTGAT-3′
PSMB4-73S	5′-CC***GAATTC***GCATGGGATCCTACGGCTCCTT-3′
PSMB4-72AS	5′-CCG***CTCGAG***TTACACCATGTCTGCGGCAAT-3′
6XHA	Gene Sequence: ***AAGCTT***ATGTACCCCTACGACGTGCCCGACTACGCCGGCTATCCTTACGATGTGCCTGATTACGCCATGGGCTACCCCTATGATGTCCCCGATTATGCCCACATGTATCCTTATGACGTCCCAGACTATGCTGGCTACCCATATGACGTGCCAGATTACGCTATGGGGTATCCATACGACGTTCCGGATTATGCC***GGATCC***

**Table 2 viruses-14-02277-t002:** Antibodies used in this study.

Name	Type	Company
Anti-beta-actin	Rabbit polyclonal Ab	Proteintech (Rosemont, IL, USA)
Anti-GFP	Mouse monoclonal Ab	Santa Cruz Biotechnology (Dallas, TX, USA)
Anti-IAV NS1	Goat polyclonal Ab	Santa Cruz Biotechnology (Dallas, TX, USA)
Anti-myc tag	Mouse monoclonal Ab	EMD Millipore Corp. (Billerica, MA, USA)
Anti-V5 tag	Mouse monoclonal Ab	Bio-Rad Laboratories Inc. (Hercules, CA, USA)
Anti-ERK2	Rabbit polyclonal Ab	Santa Cruz Biotechnology (Dallas, TX, USA)
Anti-PSMB4	Mouse monoclonal Ab	Abnova (Taipei, Taiwan)
Anti-PSMB4	Rabbit polyclonal Ab	ABclonal Inc. (Woburn, MA, USA)
Anti-PSMB4	Mouse monoclonal Ab	Santa Cruz Biotechnology (Dallas, TX, USA)
Anti-IAV M1	Mouse monoclonal Ab	Bio-Rad Laboratories Inc. (Hercules, CA, USA)
Goat anti-mouse IgG, HRP-conjugated		Perkin Elmer (Waltham, MA, USA)
Goat anti-rabbit IgG, HRP-conjugated		Perkin Elmer (Waltham, MA, USA)
Goat anti-rabbit IgG, AP-conjugated		Perkin Elmer (Waltham, MA, USA)
Mouse anti-V5 mAB, FITC-conjugated		Invitrogen (Waltham, MA, USA)
Goat anti-mouse, rhodamine-conjugated		Chemicon (Temecula, CA, USA)
Donkey anti-goat, FITC-conjugated		Santa Cruz Biotechnology (Dallas, TX, USA)
Goat anti-mouse, cy3-conjugated		Jackson lab (Bar Harbor, ME, USA)

## Data Availability

Not applicable.

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
