# Peer review of "Cellular PSMB4 Protein Suppresses Influenza A Virus Replication through Targeting NS1 Protein"

_viruses, 2022, doi:10.3390/v14102277_

Round 1

Reviewer 1 Report

In this manuscript, the authors demonstrated that cellular PSMB4 protein interacts with NS1 protein and may facilitate the degradation of NS1 protein, which in turn suppresses IAV replication. Overall, it can be an interesting study and a valuable contribution to the field. However, several points require attention and should be addressed as described below.   

1. In Fig. 1, can the authors explain why PSMB4 was selected for this study? Although the authors mentioned that " Of the 56 yeast clones harboring cDNAs encoding potential NS1 binding proteins, 25 of them contained the coding region of varied sizes for proteasome subunit beta type-4 (PSMB4) (Fig. 1). ", it is not related to Fig. 1. Also, please provide more details in the Materials and Methods for the yeast two-hybrid screening. What's the reason for splitting the PSMB4 protein into two fragments (a.a. 1-72 and 73-264)? Furthermore, please add the labeling for panel B and C as same as panel A (left).

2. In Fig. 2, did the authors simultaneously detect two proteins (PSMB4 and NS1) on a single membrane during WB? It is not recommended due to their similar molecular weights.

3. In Fig. 3, each panel should have one set of images and the rest could be removed to Supplementary material. Also, scale bar was required.

4. In Fig. 4, please remove the line on the top of panel A and the outline of images in panel B should be same as panel A, C.

5. In Fig. 5E, it clearly shows that NS1 degradation by PSMB4 could be enhanced in the present of MG132. Why the authors said "Thus, NS1 degradation by PSMB4 is likely via a MG132-independent pathway."? MG132 as a potent proteasome inhibitor reduced the degradation of PSMB4 protein. But how can MG132 promote NS1 degradation by PSMB4 since proteasome is inhibited by MG132? It is better to explain in the Discussion.

6. In Fig. 6 and 7, please remove the horizontal lines. What is the lower band of M1 in the Fig. 7C but not 7A?

7. For all tables in this manuscript, please follow three-line table format.

8. other errors to be corrected.

line 40   remove the space after "NEP".

line 48   add a space after "NS1".

line 78   please give the full name of PEI.

line 86   add a space between 1 and mM. Please check all text.

line 90   2x106 should be replaced with 2 x106

line 90   hours was used and hrs was used later (line 104), but hours occurred again (line 116). Please use the full name for the first time and the abbreviation after that. Please check all text.

line 96   4oC should be replaced with 4℃. Please check all text.

line 100 no space between 5 and %. Please check all text.

line 105 uM should be replaced with μM. Please check all text.

line 177 NS-1 should be replaced with NS1.

line 223 add a space between NS1 and protein.

line 238 remove point after (B). Please check all text.

Author Response

Reviewer 1

In this manuscript, the authors demonstrated that cellular PSMB4 protein interacts with NS1 protein and may facilitate the degradation of NS1 protein, which in turn suppresses IAV replication. Overall, it can be an interesting study and a valuable contribution to the field. However, several points require attention and should be addressed as described below.   

  1. In Fig. 1, can the authors explain why PSMB4 was selected for this study? Although the authors mentioned that " Of the 56 yeast clones harboring cDNAs encoding potential NS1 binding proteins, 25 of them contained the coding region of varied sizes for proteasome subunit beta type-4 (PSMB4) (Fig. 1). ", it is not related to Fig. 1. Also, please provide more details in the Materials and Methods for the yeast two-hybrid screening. What's the reason for splitting the PSMB4 protein into two fragments (a.a. 1-72 and 73-264)? Furthermore, please add the labeling for panel B and C as same as panel A (left).

Response: Thanks for these helpful suggestions. We revised the manuscript based on your suggestions, please refer to lines 72-82; lines 170-190. Our previous publication was cited as a new reference regarding yeast two-hybrid system. [Chee-Hing Yang, Hui-Chun Li, Cheng-Huei Hung, Shih-Yen Lo* (2015) Studying coronavirus-host protein interactions. Methods in Molecular Biology, Volume 1282, pages: 197-212.]

  1. In Fig. 2, did the authors simultaneously detect two proteins (PSMB4 and NS1) on a single membrane during WB? It is not recommended due to their similar molecular weights.

Response: Thanks for this comment. To rule out this possibility, co-IP was conducted again using PSMB4 with 6xHA tag instead of myc tag in the revised manuscript. In this case, these two proteins could be separated in the same SDS-PAGE and split into two membranes. Please refer to revised Figure 2.

  1. In Fig. 3, each panel should have one set of images and the rest could be removed to Supplementary material. Also, scale bar was required.

Response: Thanks for these suggestions. We revised the manuscript based on your suggestions, please refer to the revised Figure 3.

  1. In Fig. 4, please remove the line on the top of panel A and the outline of images in panel B should be same as panel A, C.

Response: The outline of images in panel B is now same as panel A, C. as suggested. However, I don’t understand [the line on the top of panel A]. There is no line on the top of panel A.

  1. In Fig. 5E, it clearly shows that NS1 degradation by PSMB4 could be enhanced in the present of MG132. Why the authors said "Thus, NS1 degradation by PSMB4 is likely via a MG132-independent pathway."? MG132 as a potent proteasome inhibitor reduced the degradation of PSMB4 protein. But how can MG132 promote NS1 degradation by PSMB4 since proteasome is inhibited by MG132? It is better to explain in the Discussion.

Response: Thanks for these comments. We revised the manuscript based on your suggestions, please refer to lines 313, 430-436.

  1. In Fig. 6 and 7, please remove the horizontal lines. What is the lower band of M1 in the Fig. 7C but not 7A?

Response: Thanks for these comments. We revised the manuscript based on your suggestions, please refer to Figures and figure legend 7 (words marked by the red color).

  1. For all tables in this manuscript, please follow three-line table format.

Response: I used the template provided by the Journal. I could not find [three-line table format] in the [submitted to Viruses]. Thus, I don’t know the [three-line table format]. I am sorry that I could not revise the tables as suggested.

  1. other errors to be corrected.

line 40   remove the space after "NEP".

line 48   add a space after "NS1".

line 78   please give the full name of PEI.

line 86   add a space between 1 and mM. Please check all text.

line 90   2x106 should be replaced with 2 x106

line 90   hours was used and hrs was used later (line 104), but hours occurred again (line 116). Please use the full name for the first time and the abbreviation after that. Please check all text.

line 96   4oC should be replaced with 4℃. Please check all text.

line 100 no space between 5 and %. Please check all text.

line 105 uM should be replaced with μM. Please check all text.

line 177 NS-1 should be replaced with NS1.

line 223 add a space between NS1 and protein.

line 238 remove point after (B). Please check all text.

Response: Thanks for these corrections. These typo-errors have been corrected as suggested (words marked by the red color).

Author Response

Reviewer 2

In this study, the authors demonstrate the physical interaction between PSMB4, a subunit of the 26S proteasome, and IAV NS1 (strain WSN). They have identified this interaction through a Y2H screening and confirmed it in human cells by co-ip.

Next, the authors showed that PSMB4 protein abundance/localization did not vary when cells were transiently transfected by NS1 nor when they were infected by IAV. On the contrary, Ectopic expression of PSMB4 reduced significantly NS1 expression when it was coexpressed in the cells. Furthermore, when PSMB4 was depleted by using shRNA, NS1abundance was increased. Treating cells with MG132, an inhibitor of 26S proteasome, led to and increase of PSMB4 expression as well as a decrease of NS1 expression.

NS1 is involved in IFN pathway inhibition, and also in translational enhancement of transient gene expression. Both these roles are partially suppressed by overexpression of PSMB4, in a dose-dependent manner.

Finally the authors showed that PSMB4 depletion increased viral replication and conversely its overexpression reduced virus production. PSMB4 acts thus as a Influenza virus restriction factor.

The results are clear, several could be improved and globally, I agree with authors results interpretations. Nevertheless, this manuscript should be improved.

Major points

Material and methods

Y2H technique : a lot of informations ar elacking, e.g. which yeast strain and plasmids were used, what was the method of screening : mating or transformation (both can be used when working with Clontech Y2H system). What were the reporter genes used, and if HIS3 was used which 3-AT concentration was used ? In figure 1, 3-AT is mentioned but no concentration was given. How yeast clones were recovered and selected ? How were cDNA clones identified ?

Response: Thanks for these helpful suggestions. We revised the manuscript based on your suggestions, please refer to lines 72-82. Our previous publication was cited as a new reference regarding yeast two-hybrid system. [Chee-Hing Yang, Hui-Chun Li, Cheng-Huei Hung, Shih-Yen Lo* (2015) Studying coronavirus-host protein interactions. Methods in Molecular Biology, Volume 1282, pages: 197-212.]

Results

Figure 1

- Y2H : out of 56 yeast clones, 25 contained the coding region of PSMB4 of varied sizes. What about the 31 others clones ? What did they mean when they talked about different sizes of the PSMB4 clones ?

- Y2H deletion mapping : albeit I understand why authors selected the 2 domains of NS1 (RBD and the rest of NS1, not exactly the ED domain was tested), I don’t understand the rationale of testing N-term part of PSMB4 (aa 1-72) and C-term part (aa 73-264). There is no information concerning that.

Response: Thanks for these helpful suggestions. We revised the manuscript based on your suggestions, please refer to lines 170-190, 196-197.

Figure 2

Why did the authors performed a co-revelation of the co-ip ? It would be far more clear with two different revelations (one with myc antibody and another with with V5 antibody). The authors should let migrate longer the gel such that the proteins would be better separated. How much protein extracts were used for each co-ip ? Did the authors performed a protein concentration assay of their cell extracts before ip ? They should engage the same amount of protein for each condition.

The co-ip is not very convincing since a faint band of myc-PSMB4 is visible in the ip in the lane in absence of V5-NS1.

Response: Thanks for this comment. Co-IP was conducted again using PSMB4 with 6xHA tag instead of myc tag in the revised manuscript. In this case, these two proteins could be separated in the same SDS-PAGE and split into two membranes. Please refer to revised Figure 2.

Figure 3

The authors should add a scale bar on the picutres. I don’t understand why several pictures are represented. They authors should include a quantitative analysis of colocalization signal.

Response: Thanks for the comments. Only one picture was presented in the revised Figure 3 and a scale bar is included as suggested. [a quantitative analysis of colocalization signal] is a good suggestion. However, this experiment was conducted several years ago. We could not quantitate the colocalization signal now.

To my opinion the authors should perform PSMB4 and NS1 subcellular localization upon IAV infection at different time points. The authors possess antibodies raised against endogenous PSMB4, as well as against NS1, they could detect PSMB4 and NS1 in infected cells. It will be far more interesting. It is well known that exogenous expression of proteins do not mimick their exact subcellular localization.

Response: I do agree with this comment! We did intend to do this experiment at the beginning. However, commercially available PSMB-4 antibodies are not good for the staining. Several commercially available antibodies listed in Table 2 were tested.

Figure 4

In this figure (and beyond), the authors performed quantification of the signals obtained on western blots. Did they performed such quantification in triplicate or only on one western-blot ? Tehy should clearly mention it in the legend of the figures. To my opinion, such a protein quantification requires biological replicates.

Response: Every experiments on WB in this manuscript have been repeated at least two times. Please refer to lines 261-262.

Figure 5

MG132 is not explained in the methods section. In legend of figure 5 we learn that they are using 10 μM of MG132 but it is not written for how long cells were treated with this drug. How do the authors explain the difference in HA-PSMB4 protein expression difference between fig 5A and 5B (without MG132) ? The presence of NS1 increase PSMB4 expression.

Response: Thanks. Please refer to lines 177-179 [words marked in red color].

What is the hypothesis of the authors to explain why NS1 is less stable in presence of MG132, the inhibitor of 26S proteasome, the cell machinery rightly required for protein degradation ?

Response: Thanks. Please refer to lines 432-438 [words marked in red color].

Figure 7

I’m quite disappointed on Fig7A, concernint M1 detection on SCR shRNA treated cells infected at MOI of 0.01 48h after infection. Very few (if there is smething detectected) M1 is detected. As well for Fig 7B, it is very difficult to believe the graphs : upon depletion of PSMB4 (very partial, see fig 5D), the authors observed more than 50 times more Pfu ? Actually I’m more confident on results concerning MOI of 3, because authors can detect M1 in SCR treated cells (24h post infection) and they observe a significant difference in Pfu but 3 times more plaques were observed, not 50 times more.

Response: Thanks for the comment. I don’t know why very few M1 is detected at MOI of 0.01 48h after infection. One possibility is that the A549 cells but not MDCK cells were used in this experiment. The viral titer generated from A549 cells is far less than that from MDCK cells [Ueda, M.; Yamate, M.; Du, A.; Daidoji, T.; Okuno, Y.; Ikuta, K.; Nakaya, T., Maturation efficiency of viral glycoproteins in the ER impacts the production of influenza A virus. Virus Res 2008, 136, (1-2), 91-7.]

For Fig 7E, the authors should explain the timing of the experiment : when was added MG132 (before infection, after, for how long, etc…) ?

Response: Thanks for the suggestion. The information is added as suggested. Please refer to lines 391-392.

Minor points

Figure 6

Some frames are appearing arounf some panels of the figure, remove them.

The authors should describe their hypothesis explaining why NS1 expression is reduced in MG132-treated cells.

Response: Thanks. Please refer to lines 432-438 [words marked in red color].

Reviewer 3 Report

Chee-Hing and colleagues found a protein, PSMB4, that interact with influenza NS1 protein in yeast two-hybrid system. And co-IP assay was used to verify the interaction of the two proteins. Notably, PSMB4 may degraded NS1 not through the ubiquitination-proteasome pathway. The effect of PSMB4 to NS1 and influenza virus were investigated as well. The PSMB4 was found suppressing the function of NS1 and inhibit the replication of influenza virus. This study seems very straight forward. A protein was found, and important functions were developed. However, to this kind of study, an important issue was missing, which is the scientific question. Without a good scientific question, we don’t know the aim of the study. But in this study, an anti-viral effect was found with PSMB4, which still worth studying.

Major revision

1.     Please explain the scientific question of the study.

2.     The PSMB4 is identified in a human fetal liver cDNA library, which is not the target of IAV. The distribution of PSMB4 is unknown. If it is not existed in lung epithelial virus, what is the aim of the study? Thus, please add the PSMB4 distribution in different organs.

3.     In figure 7, virus titers were shown in fold change. Please do a multi-step growth curve and show virus titers not fold-change. And virus titers in different time points need to be shown.

4.     Without PSMB4 knockout cells, it is hard to explain the function of it to IAV. Although authors claim that PSMB4 knockout cells are unable to be obtained.

5.     In figure 7A, 7C, M1 concentrations were also affected by PSMB4. Thus, the inhibition of virus replication is not surely because the degradation of NS1 protein. Did you investigate if PSMB4 has any effect on other proteins of IAV? And please discuss it in the manuscript.

6.     Please explain the biological significance of this study. PSMB4 suppresses influenza virus replication in vitro. What do you expect in vivo? What do you want to do with PSMB4? In future study, is it worth to explore the mechanism how PSMB4 degraded NS1 and why?

Author Response

Reviewer 3

Chee-Hing and colleagues found a protein, PSMB4, that interact with influenza NS1 protein in yeast two-hybrid system. And co-IP assay was used to verify the interaction of the two proteins. Notably, PSMB4 may degraded NS1 not through the ubiquitination-proteasome pathway. The effect of PSMB4 to NS1 and influenza virus were investigated as well. The PSMB4 was found suppressing the function of NS1 and inhibit the replication of influenza virus. This study seems very straight forward. A protein was found, and important functions were developed. However, to this kind of study, an important issue was missing, which is the scientific question. Without a good scientific question, we don’t know the aim of the study. But in this study, an anti-viral effect was found with PSMB4, which still worth studying.

  1. Please explain the scientific question of the study.

Response: Indeed, scientific question is a big issue. In this study, PSMB4 was identified as a restriction factor for IAV. The detailed mechanisms regarding the facilitation of NS1 reduction by PSMB4 in the presence of MG132 required further investigations. If the mechanism of NS1 degradation is uncovered, it will help to find a way to degrade NS1 and in turn suppress IAV replication.

  1. The PSMB4 is identified in a human fetal liver cDNA library, which is not the target of IAV. The distribution of PSMB4 is unknown. If it is not existed in lung epithelial virus, what is the aim of the study? Thus, please add the PSMB4 distribution in different organs.

Response: Thanks for this comment. The reason why human fetal liver cDNA library was used was added in lines 173-174. The distribution of PSMB4 in the tissues was added in lines 65-67.

  1. In figure 7, virus titers were shown in fold change. Please do a multi-step growth curve and show virus titers not fold-change. And virus titers in different time points need to be shown.

Response: Thanks for this comment. Virus titers not fold-change were determined as suggested in revised figure 7.

  1. Without PSMB4 knockout cells, it is hard to explain the function of it to IAV. Although authors claim that PSMB4 knockout cells are unable to be obtained.

Response: Indeed, the effect of PSMB4 on IAV should be more obvious using PSMB4 knockout cells than using knockdown cells. However, in this study, effect of PSMB4 on IAV could be studied in knockdown cells (Fig. 7).

  1. In figure 7A, 7C, M1 concentrations were also affected by PSMB4. Thus, the inhibition of virus replication is not surely because the degradation of NS1 protein. Did you investigate if PSMB4 has any effect on other proteins of IAV? And please discuss it in the manuscript.

Response: I believe that there is a misunderstanding here. To prove that M1 is not affected by PSMB4 directly, an experiment was conducted in revised Fig. 5B. In figure 7A, 7C, M1 concentrations were affected by PSMB4 indirectly. That is to say, PSMB4 reduced the NS1 amount, which is essential for IAV replication. In turn, less viral titer generates fewer M1 protein.

  1. Please explain the biological significance of this study. PSMB4 suppresses influenza virus replication in vitro. What do you expect in vivo? What do you want to do with PSMB4? In future study, is it worth to explore the mechanism how PSMB4 degraded NS1 and why?

Response: In this study, PSMB4 was identified as a restriction factor for IAV. The detailed mechanisms regarding the facilitation of NS1 reduction by PSMB4 in the presence of MG132 required further investigations. If the mechanism of NS1 degradation is uncovered, it will help to find a way to degrade NS1 and in turn suppress IAV replication. Yes, I believe that it is worth to explore the mechanism how PSMB4 degraded NS1. The degradation of NS1 (i.e., facilitated by MG132) is different from that of most proteins (i.e., prevented by MG132). We may find a novel pathway for protein degradation.

Round 2

Author Response

- In manuscript version#1, authors talked about clones containing the coding region of varied sizes for PSMB4. Now, the authors tell that 25 clones are containing the coding region of PSMB4, but there is no more info about clones of different sizes.

I asked the rationale of working with PSMB4 fragments 1-72 and 73-264 ; the authors simply added they cut PSMB4 artificially but without explaining why. Is there any link with the different clones obtained in the primary Y2H screen ?

Response: Thanks for your comments. Indeed, it is due to the different clones obtained in the primary Y2H screen. Please refer to the section [Cellular PSMB4 was found to interact with IAV NS1 protein using yeast two-hybrid system] (marked in red color).

- Material and methods concerning yeast two-hybrid system. The authors improved a lot the method section. They could mention that positive colonies growing on -W-H-L we streaked on -W-L-H+3AT (100 mM) and only growing colonies on that selection medium were selected.

Response: Thanks for your suggestions. The sentence has been added in [Materials and methods], marked in red letters.

- Figure 1 legend: yeast culture medium is not YEPD, but YNP.

Response: Thanks for the corrections. YNP (marked in red letters) but not YEPD was used in Figure 1 legend.

Reviewer 3 Report

The questions have been well addressed; the scientific significance is better elucidated; and the details are better defined. I agree the manuscript to be published.

Author Response

Thanks for your helpful suggestions.